# Psychotherapeutic Interventions for Depressive Symptoms in Community-Dwelling Older Adults: A Systematic Review with Meta-Analysis

**DOI:** 10.3390/healthcare12242551

**Published:** 2024-12-18

**Authors:** Bruno Morgado, Celso Silva, Inês Agostinho, Filipe Brás, Pedro Amaro, Leonel Lusquinhos, Maria Revés Silva, Cesar Fonseca, Núria Albacar-Riobóo, Lara Guedes de Pinho

**Affiliations:** 1Escola de Doctorat, Universitat Rovira y Virgili, 43005 Tarragona, Spain; nuria.albacar@urv.cat; 2Higher School of Health, Polytechnic University of Portalegre, 7300-110 Portalegre, Portugal; pedro.amaro@ipportalegre.pt; 3Higher School of Health, Polytechnic Institute of Beja, 7800-295 Beja, Portugal; celsosilva30@gmail.com; 4LA-REAL, Comprehensive Health Research Centre (CHRC), University of Évora, 7000-811 Évora, Portugal; maria.joao.silva@uevora.pt (M.R.S.); cfonseca@uevora.pt (C.F.); lmgp@uevora.pt (L.G.d.P.); 5Lisbon West Local Health Unit, 2770-219 Lisbon, Portugal; ines_neca@hotmail.com; 6Alto Alentejo Local Health Unit, 7300-853 Portalegre, Portugal; filipe.bras.94@gmail.com; 7Nursing Department, University of Évora, 7000-811 Évora, Portugal; leonel.oliveira@uevora.pt

**Keywords:** psychotherapies, non-pharmacological therapy, non-pharmacological treatment, depression, older adults, community setting, geriatric psychiatry, geriatric psychology

## Abstract

The global ageing population faces rising depression rates due to social, economic, and health changes. Depression in older adults, often linked to isolation and health issues, requires comprehensive care. Psychotherapeutic interventions could be effective in reducing symptoms, offering personalized and holistic support. Particularly low-threshold interventions, such as those offered in community-dwelling older adults, which older adults can easily access and which may reduce stigma, promise to close the treatment gap. This review examines community-based psychotherapeutic interventions for older adults with depression. Methods: This review investigates psychotherapeutic interventions for reducing depressive symptoms in older adults in a community setting. RCTs were assessed using Joanna Briggs Institute tools. The following databases were searched: CINAHL Plus with Full Text, MedicLatina, MEDLINE with Full Text, and the Psychology and Behavioral Sciences Collection. Results: A meta-analysis of 13 studies with 1528 participants showed a medium, significant pooled effect size at post-intervention (Hedges’ *g* = −0.36, *p* < 0.001) and substantial heterogeneity. Follow-up analysis of studies indicated a small, non-significant effect (Hedges’ *g* = −0.17, *p* = 0.27). Group interventions, particularly the “reminiscence protocol”, had the largest significant effect. Discussion: This systematic review and meta-analysis found that in community-dwelling older adults’ group psychotherapeutic interventions, particularly the “reminiscence protocol” and “modified behavioral activation treatment” are most effective for reducing depressive symptoms. Individual psychotherapeutic interventions like “prevention of suicide in primary care elderly” and “behavioral activation” also show effectiveness, with group psychotherapeutic interventions being generally more effective than when these treatments are offered in individual psychotherapeutic interventions. Conclusion: Group and individual psychotherapeutic interventions reduce depressive symptoms in community-dwelling older adults, with group psychotherapeutic interventions being more effective.

## 1. Introduction

As described in [1], population aging is an increasing trend worldwide, with the World Health Organization (WHO) projecting that by 2050, the number of people aged 60 and older will triple, rising from 1 billion in 2020 to 3 billion [1]. This growth is driven by improvements in health, nutrition, and access to medical services [1]. According to the UN report from 2019, about 16% of the global population will be aged 65 and older by 2050, compared to 9% in 2019. These figures highlight the urgent need for policies that address the social and economic challenges of aging [1]. Several factors contribute to this concerning trend, including social, economic, and health-related changes [2].

Depression among older adults is a serious and multifaceted issue, with various underlying causes. Studies indicate that the prevalence of depressive symptoms in individuals aged 60 and older ranges from 10% to 15%, potentially rising to 20% in more vulnerable groups [3]. Furthermore, a systematic review published in 2020 revealed that depression is one of the most common mental health conditions in this age group, often exacerbated by factors such as social isolation, chronic illnesses, and the loss of loved ones [3,4]. These findings highlight the need for targeted interventions for the mental health of older adults. In addition to the physical and biological changes associated with ageing, such as cognitive decline and deteriorating physical health, seniors often face significant emotional and social challenges [5,6]. Social isolation, loss of loved ones, financial problems, health concerns, and declining independence are among the factors that contribute to depression among older adults [7] Furthermore, depressive symptoms frequently coexist with other medical conditions (e.g., diabetes, high blood pressure), rendering diagnosis and treatment even more challenging. Furthermore, issues related to the stigma surrounding mental health in this population can exacerbate the problem, leading to underreporting and undertreatment of depression [5,6,8].

Although international guidelines recommend psychotherapeutic interventions in addition to antidepressants for the treatment [8] of depressive symptoms [9], many older adults never receive this treatment. Psychotherapeutic interventions offer personalized attention tailored to the unique needs and circumstances of patients. Integrating psychotherapeutic interventions into holistic geriatric care models can enhance outcomes and improve overall quality of life for older adults [10]. By incorporating psychotherapeutic interventions alongside medical management, social support services, and lifestyle interventions, healthcare professionals can provide comprehensive and integrated care that addresses the multifaceted needs of older adults [10,11]. To improve dissemination and integration into existing comprehensive care programs, low threshold psychotherapeutic interventions have been developed that can be administered by non-specialized healthcare professionals [12].

Care for people with depressive symptoms should be dynamic, flexible, and participatory, responding to their specific needs by establishing individual objectives and shared decision making [13]. One of the primary advantages of psychotherapeutic interventions lies in their adaptability to the individual needs of older adults [14]. Various psychotherapeutic interventions modalities, such as cognitive behavioral therapy (CBT), interpersonal therapy (IPT), and group psychotherapeutic interventions, can be tailored to meet the preferences and capacities of older adults. For instance, CBT can assist older adults in identifying and modifying negative thought patterns, while IPT focuses on enhancing interpersonal relationships and social support [15,16]. Additionally, group psychotherapeutic interventions provide a valuable opportunity for shared experiences, which can reduce isolation and strengthen community bonds [16]. Another crucial aspect of the efficacy of psychotherapeutic interventions is their ability to address specific risk and protective factors associated with depression among older adults [17]. For example, loneliness and social isolation are significant risk factors for depression in this age group, and psychotherapeutic interventions can help mitigate these negative effects by promoting social connection and mutual support [14,15,18]. Moreover, psychotherapeutic interventions can aid in adapting to stressful life events, such as retirement, spousal loss, or health problems, thereby reducing the emotional impact of these transitions [19].

The efficacy of psychotherapeutic interventions in depressive symptoms among older adults is supported by a growing body of scientific evidence [20]. Studies consistently demonstrate that psychotherapeutic interventions are effective in reducing depressive symptoms, preventing relapse, and enhancing psychological well-being in older adults [21]. Furthermore, psychotherapeutic interventions can be as effective as, or even more effective than, antidepressant medication in certain cases, with the added advantage of fewer side effects and associated risks [22].

Examining depressive symptoms interventions in community-dwelling older adults is essential as it aims to improve accessibility for individuals. Community-dwelling, such as churches, nursing homes, and community centers, provide an opportunity to reduce barriers, including the stigma often associated with mental health treatment. Additionally, these interventions tend to be more cost-effective, empowering, and engaging for individuals within the community. By fostering stronger connections with local networks, these programs can offer better continuity of care, allowing for a more holistic and preventative approach to managing depressive symptoms.

A preliminary search of the PubMed database using the Boolean equation (“depression AND non-pharmacological interventions AND older adults AND community AND systematic review”) yielded 25 results. None of these studies focused exclusively on psychotherapeutic interventions for community-dwelling older adults with depressive symptoms.

Our review aims to bridge this gap by exclusively examining community-dwelling older adult-based psychotherapeutic interventions, thus contributing valuable, previously unexplored knowledge to the field.

In this systematic review with meta-analysis, the aim is to test psychotherapeutic interventions that reduce depressive symptoms in community-dwelling older adults.

## 2. Methods

### 2.1. Registration

The protocol for this review has been registered with PROSPERO (CRD42023449190). When we initially developed the protocol, we didn’t consider carrying out a meta-analysis, but to quantify the combined results of the included studies, increase statistical power, and reduce the uncertainty of the findings, we decided to go ahead with it.

### 2.2. Review Question

This review will be conducted to answer the following PICO question:

What psychotherapeutic interventions (I) contribute to the reduction of depressive symptoms (O) in community-dwelling older adults with diagnosed depression (P), compared to no treatment or different types of interventions (C)?

### 2.3. Eligibility Criteria

The inclusion criteria were (1) RCT studies examining (2) psychotherapeutic interventions applied among (3) community-dwelling (4) older adults aged 60 or over (5) with a diagnosis of depression (6) published between 2013 and 2023. Studies examining participants that included individuals with co-morbid psychological disorders were not included in the analysis. The justification for including RCTs in this literature review stems from the rigorous methodological standards that RCTs uphold, which are considered the gold standard for evaluating the effectiveness of interventions [22]. RCTs offer the highest level of evidence due to their ability to minimize bias through randomization and the use of control groups. This methodological strength allows for a clearer determination of causality between the intervention and the observed outcomes, thereby providing more reliable and valid results. Excluding other study designs, such as observational studies, case-control studies, and cohort studies, strengthens the validity of the review’s findings. While these designs can offer valuable insights, they are more susceptible to various biases, such as selection bias, recall bias, and confounding variables, which can compromise the accuracy of the results. By focusing exclusively on RCTs, our review aims to ensure that the conclusions drawn are based on the most robust and credible evidence available. The specificity of our study’s scope is defined by its aim to identify trials on any psychotherapeutic interventions that effectively reduce or attempted to reduce depressive symptoms in community-dwelling older adults. This focus is crucial because older adults often face unique psychological, social, and physical challenges that can influence the manifestation and treatment of depressive symptoms. By narrowing our scope to psychotherapeutic interventions, we intend to comprehensively evaluate the effectiveness of various therapeutic approaches, such as cognitive behavioral therapy, interpersonal therapy, and mindfulness-based interventions, specifically within this demographic. Furthermore, our study’s emphasis on community-dwelling older adults highlights an often-overlooked population in existing research. While many studies target institutionalized older adults or those with significant dependencies, our review seeks to address the gap in understanding the impact of psychotherapeutic interventions on those living in community-dwelling. This distinction is vital, as community-dwelling older adults (e.g., which would also include nursing homes, churches) may have different support systems, levels of autonomy, and social interactions, all of which can influence the effectiveness and feasibility of psychotherapeutic interventions.

### 2.4. Search Strategy

In this review, a comprehensive literature search was developed, and the databases consulted were: CINAHLPlus with Full Text, MedicLatina, MEDLINE with FullText, and the Psychology and Behavioral Sciences Collection. Studies published in English, Portuguese, Spanish, or German were considered. The research included the combinations of search terms according to the Medical Subject Headings (MeSH) terms:

(“Depression”) OR (“depressive disorder”) OR (“depressive symptoms”) OR (“major depressive disorder”) AND (“psychotherapy”) OR (“cognitive behavioral therapy”) OR (“psychotherapy groups”) OR (“psychotherapy, brief”) OR (“behavior therapy”) OR (“non-pharmacological”) OR (“non-pharmacological interventions”) OR (“non-pharmacological treatment”) AND (“community”) OR (“home care”) OR (“home care services”) OR (“home health care”) OR (“home healthcare”) OR (“home nursing”) AND (“older adult”) OR (“elderly”) OR (“geriatric patients”) (See Appendix A).

### 2.5. Data Collection and Analysis

Study selection included various steps. The studies resulting from searches in each database were exported into Mendeley and duplicates were removed. To minimize bias, two reviewers independently assessed the inclusion of the studies by reading the titles, abstracts, and keywords, excluding those that did not fit the inclusion criteria (Figure 1—PRISMA flowchart). A third reviewer was consulted in case of disagreement or doubts. Subsequently, we proceeded to the full-text assessment phase using the same method.

### 2.6. Data Extraction

All RCT psychotherapeutic interventions that reduce depressive symptoms in community-dwelling older adults (i.e., over 60 years of age) were included. Duplicates were removed and then all identified studies were reviewed by four of the co-authors who provided a final decision on inclusion. The results of the analyses were then cross-referenced, and in the event of disagreement about the inclusion of a particular article, a fifth author carried out the tie-breaker analysis based on the quality of the article and the inclusion criteria. Reviewers identified the articles for inclusion by reading the titles, abstracts, and full manuscripts, and chose articles fulfilling the inclusion criteria (RCTs including a comparison of the effectiveness of a psychotherapeutic intervention compared to treatment as usual, waitlist control group or active control group was provided, population at least 60 years of age, community-dwelling). Information on the psychotherapeutic intervention and the clinical assessment instruments used to assess depressive symptoms were extracted into the summary tables below. We included both self-report and clinician-rated measures of depressive symptoms in the study. In cases where both types of measures were utilized, we specified which one was used for the analysis, ensuring the consistency and relevance of the data interpreted for the research objectives.

### 2.7. Strategy for Data Synthesis and Analysis

The data extracted from the RCTs were analyzed using IBM^®^ SPSS^®^ Statistics V28 software. A random effects model was used, based on the assumption of heterogeneity of the studies collected, generating more conservative estimates of precision, with weighting based on the inverse variance method in which studies with lower variance (more precise) are given greater weight [23]. To assess the effect of non-pharmacological interventions on the study population, the standardized difference in means was used as a measure of effect (Hedges adjusted *g*), with standard error adjustment. This adjustment allows greater weight to be given to studies with lower standard error in the meta-analysis. Hedges’ *g* provides a more conservative estimate, useful in small samples, with an effect interpretation: 0.2—small; 0.5—medium; 0.8—large [23]. To quantify the degree of variability in the whole sample and in the subgroups, homogeneity estimates were calculated. A statistically significant value (*p* < 0.05) suggests that there is heterogeneity. I^2^ estimates represent the proportion of variability between studies that is not due to chance (random). The magnitude of I^2^ has been interpreted as: 0% to 40% may not be important; 30% to 60% may represent moderate heterogeneity; 50% to 90% may represent substantial heterogeneity; 75 to 100% may represent considerable heterogeneity [21]. The significance of the observed I^2^ value was assessed using the *Q* statistic (chi^2^), where a *p*-value of <0.05 suggests homogeneity. Publication bias was assessed using the Egger test. In the measurement instruments used in the studies, higher scores (mean) suggest greater depressive symptomatology. Subgroup analysis (group vs. individual non-pharmacological intervention) will be carried out to investigate the effects on the outcome. Subgroup analysis allows for a more robust analysis of the relationship between the intervention and the results obtained [10].

### 2.8. Quality of the Evidence and Risk of Bias

As this is a review of RCTs, the Joanna Briggs Institute quality assessment tool was applied for each selected article to help assess the reliability, relevance, and results of the published articles [24]. To minimize bias, four reviewers independently assessed the inclusion of studies by reading the title, abstracts, and keywords and excluded those that did not meet the inclusion criteria for this review. The fifth reviewer was consulted in case of disagreement and provided a decision based on the forementioned criteria. Subsequently, the full texts were assessed by the ten researchers for a rigorous analysis, so that we could draw the best and most accurate conclusions. For an overview of the review process see the PRISMA flowchart (Figure 1).

## 3. Results

### 3.1. Study Characteristics

To facilitate the analysis and discussion of the results, Table 1 presents a summary of study characteristics with the available data from each study, organized by author, year, sample, aims, results, conclusions, and country.

This systematic review includes studies conducted across a range of countries, with the majority based in the USA (6 studies), followed by China, Thailand, Switzerland, Hong Kong, and the Netherlands. The studies primarily involved older adults (generally 60+ years). The populations studied were diverse, including older adults in rural areas, residential care, and nursing homes, as well as those who were community-dwelling or left behind by migrating family members.

The studies yielded several key findings regarding the effectiveness of various interventions in reducing depressive symptoms, anxiety, and other associated conditions in older adults:

Reminiscence therapy: In a USA-based study, structured group reminiscence therapy significantly reduced depression and anxiety among older female hookah users (*p* < 0.05). It also reduced the number of cigarettes smoked daily, demonstrating a dual benefit for mental health and smoking cessation.

Behavioral activation (BA): A study conducted in Thailand found that BA was effective in improving heart rate variability (HRV), which is linked to depression reduction in older adults with subthreshold depressive symptoms. A similar BA intervention in China reduced depressive symptoms in rural left-behind older adults compared to regular care.

Cognitive behavioral therapy (CBT): Multiple studies from the USA highlighted the efficacy of CBT-based interventions. An eight-week internet-supported CBT program in Switzerland was effective for some older adults, while a study on cognitive behavioral therapy for insomnia (CBT-I) in the USA prevented major depressive disorder in older adults with insomnia. Additionally, telephone CBT-I in older adults with osteoarthritis pain improved sleep, fatigue, and pain management.

Mindfulness-based cognitive therapy (MBCT): A Hong Kong study showed that MBCT was effective in reducing depressive symptoms and improving mindfulness and rumination. This was particularly beneficial for older adults with more severe depression and dysfunctional cognition.

Music therapy: In a USA-based study, interactive group music therapy was found to decrease depressive symptoms more effectively than recreational group singing among older nursing home residents.

Autobiographical memory interventions: Studies from the USA and the Netherlands explored the impact of autobiographical memory interventions. While both studies reported improvements in depressive symptoms, the interventions did not differ significantly from control conditions. Specific memory retrieval was found to correlate with faster reductions in depression.

Prevention of suicide in primary care (PROSPECT): A large-scale study in the USA examined the effect of the PROSPECT intervention on suicidal ideation and depression in older adults. The intervention led to a more rapid decrease in suicidal ideation and a more favorable reduction in depressive symptoms compared to usual care, especially among those with suicidal ideation.

In Table 2, the results regarding the psychometric instruments used to assess levels of depressive symptoms are presented. It is observed that only seven studies used depression scales, revealing a significant heterogeneity among the instruments employed, and not all of them are specifically designed for the geriatric population.

The included studies in this meta-analysis employed a variety of instruments to assess depression, reflecting the diversity in methodological approaches and target populations. Among the commonly used tools were the Patient Health Questionnaire-9 (PHQ-9) and the Montgomery Åsberg Depression Rating Scale—Self Rated, both of which are well-established for their reliability in detecting depressive symptoms in clinical and general populations. Additionally, tools tailored to older adults were frequently utilized, such as the Geriatric Depression Scale (GDS) and the Elderly Depression Inventory, which are designed to capture depressive symptoms specific to this demographic. One study included a culturally adapted instrument, the 30 self-rated items of the Thai Geriatric Depression Scale, underscoring the importance of local context in the assessment of mental health.

Furthermore, the Hamilton Depression Rating Scale (HDRS), a clinician-administered instrument, was employed in some studies, providing a contrast to the self-reported measures by offering an objective evaluation of depressive severity. This variation in instruments highlights the heterogeneity in study designs and populations, which should be considered when interpreting the meta-analytic findings. A more detailed examination of these tools within each study would enhance the understanding of how depression was measured, contextualizing the results and their implications.

### 3.2. Meta-Analysis

A total of 13 studies (15 comparisons) were submitted to the meta-analysis involving 1528 participants [23,25,26,27,28,29,30,31,32,33,34,35,36].

At post-intervention, the pooled effect size was medium and statistically significant (Hedges’ adjusted *g* = −0.36, Z = 3.53, *p* < 0.001) with substantial heterogeneity (I^2^ = 69%, chi^2^ = 36.08, *p* < 0.001) (Figure 2).

A meta-analysis was also carried out which included eight studies (nine comparisons) with the measure of effect at follow-up [23,25,26,27,28,29,30,31,32,33,34,35,36]. The results indicated a small non-significant combined effect (Hedges’ adjusted *g* = −0.17, Z = −1.10, *p* = 0.27), with considerable heterogeneity (I^2^ = 82%, chi^2^ = 27.44, *p* < 0.001). Due to the substantial heterogeneity, we did not use the pooled effect size, opting instead to explore the effect sizes of individual studies and subgroup analyses (Figure 3). At post-intervention, there was a significant average combined effect for group non-pharmacological interventions (Hedges’ adjusted *g* = −0.48, *p* < 0.001) and a small non-significant combined effect for individual interventions (Hedges’ adjusted *g* = −0.15, *p* = 0.40). There was evidence of homogeneity between subgroups (chi^2^ = 2.515, *p* = 0.113). The group intervention with the largest significant effect (large effect) was the “reminiscence protocol” (Hedges’ adjusted *g* = −1.79, *p* < 0.001, [−2.76, −0.81]. The individual intervention with the largest significant effect (medium effect) was the “prevention of suicide in primary care elderly” (Hedges’ adjusted *g* = −0.30, *p* < 0.001, [−0.48, −0.12] (Figure 4).

The coefficients of the intercepts for all categories (group, individual, and general intervention) are not statistically significant (*p* > 0.05), so there is no evidence of publication bias (Figure 5).

#### 3.2.1. Assessment of the Risk of Bias

The evaluation components of the JBI critical appraisal checklist for RCTs are random allocation of participants, blinding of allocation, initial similarity between groups, blinding of participants and investigators, blinding of outcome assessors, reliability and validity of outcome measures, adequacy of follow-up and management of losses, intention-to-treat (ITT) analysis, sufficiency of follow-up period, and use of appropriate statistical methods. The final classification regarding the risk of bias was based on the percentage of “unclear” and “no” attributed to each study for the total of questions evaluating the article. Finally, they were classified as low risk of bias (<33.33%), unclear risk of bias (between 33.33% and 66.66%), and high risk of bias if the score exceeded 66.66%.

Overall, the evaluation of the studies selected for analysis shows a low risk of bias (Figure 6).

#### 3.2.2. Subgroup Analyses

In addition to the subgroup analysis already conducted (effects of individual and group non-pharmacological interventions), if we compare studies conducted in the USA vs. non-USA settings, interventions conducted in the USA (Hedges’ adjusted *g* = −0.34, Z = −3.53, *p* < 0.001) with low heterogeneity (I^2^ = 28.9%) showed a small effect similar to interventions conducted in other countries (Hedges’ adjusted *g* = −0.35, Z = −3.53, *p* < 0.001) with substantial heterogeneity (I^2^ = 75.7%) (Figure 7).

#### 3.2.3. Sensitivity Analysis

Sensitivity analysis was developed taking into account the elimination of potential outliers, using the difference between the third quartile and the first quartile of results with a factor of 1.5. 2 studies were removed (Bazrafshan et al., 2021 [23]; Ayudhaya et al., 2022 [25]). There was a reduction in heterogeneity (I^2^ = 0%) with a small effect (Hedges’ adjusted *g* = −0.35, Z = −6.72, *p* < 0.001).

#### 3.2.4. Moderator Analysis

A set of univariate and multivariate random effects meta-regression analyses was carried out with the aim of understanding how the covariates under study can explain the residual heterogeneity in the results of the intervention effects. Of the covariates tested, “country” showed an impact on residual heterogeneity (*p* < 0.001, I^2^ = 0.1, R^2^ 100%). Multivariate analyses were not statistically significant (Table 3).

## 4. Discussion

This systematic review with meta-analysis aims to test psychotherapeutic interventions that reduce depressive symptoms in community-dwelling older adults.

According to our results, the psychotherapeutic interventions in community-dwelling older adults with the largest effects for reducing depressive symptoms are the “reminiscence protocol” [23] and “modified behavioral activation treatment (MBAT)” [26]. Meanwhile, the results of individual psychotherapeutic interventions in community-dwelling older adults indicate that the most effective in reducing depressive symptoms are the “prevention of suicide in primary care elderly” [35] and “behavioral activation (BA)” [24].

The reminiscence protocol proved effective in promoting cognitive stimulation and emotional regulation [23]. Recalling positive memories activates brain areas related to memory and emotional processing, helping to maintain or improve cognitive functions and slow cognitive decline [23], as other studies add that the practice of reminiscence promotes the release of neurotransmitters such as dopamine, which is associated with feelings of pleasure and reward, contributing to mood improvement [23]. A pilot study carried out in Portugal using this technique with community dwelling older adults, specifically in a nursing home during the pandemic, found that the results in terms of experiencing positive affections, which are important for improving mood, are very much related to the opportunity to interact, rather than remaining alone. The sharing that takes place between participants stimulates friendship and understanding between them, giving them a sense of acceptance in the group. Through the learning that comes from sharing other lives, participants realize that each life is unique and interesting, even if some seem sad or frustrated. The group itself builds a sense of belonging and cohesion among them that helps them overcome feelings of loneliness. In addition, memories come into play here as a principle of this therapeutic intervention to also help them validate their own “self” [35].

MBAT has also proved effective by combining behavioral activation techniques with specific modifications for older adults [25]. This type of intervention in community-dwelling older adults aims to increase participation in meaningful activities, which improves mood and reduces depressive symptoms by reinforcing positive behaviors and increasing physical activity [25]. Several studies indicate that behavioral activation is associated with neurophysiological changes, such as increased levels of neurotransmitters like serotonin and dopamine, which are often imbalanced in individuals with depressive symptoms [24,25]. It should be noted that the two interventions with the largest effect size involve changes in neurotransmitters, which may be one of the reasons for the improvement in symptoms.

For individual interventions, the prevention of suicide in primary care elderly (PROSPECT) program stands out for its effectiveness in reducing depressive symptoms in community-dwelling older adults [35]. This program focuses on the early identification and treatment of depressive symptoms in older adults, using an integrated approach that includes physician education and intensive case management [34]. PROSPECT addresses both psychosocial and physiological factors, improving adherence to treatment and providing ongoing support, which can improve neuroplasticity and reduce depressive symptoms [34].

Behavioral activation (BA) in individual community-dwelling older adults’ contexts has also been effective in encouraging older adults to engage in pleasurable and meaningful activities [25]. This method relieves the symptoms of depression by promoting the release of neurotransmitters such as dopamine, which is crucial for mood regulation [22]. Participation in meaningful and pleasurable activities can lead to positive neurophysiological changes, resulting in improvements in mood and cognition [24].

These results suggest that both group and individual interventions can be effective, with specific approaches being more appropriate depending on the therapeutic context and the individual needs of the older adults. However, in this review, group psychotherapeutic interventions demonstrated greater effect sizes than individual interventions. These findings have been confirmed by other authors, indicating that group psychotherapy provides a supportive environment where individuals can connect with peers facing similar challenges [37]. Through group discussions, shared experiences, and mutual support, participants can gain insights, learn coping strategies, and develop a sense of belonging [38,39]. Group interventions also offer opportunities for socialization and combatting feelings of loneliness and isolation, which are common contributors to depressive symptoms among the older adult [37].

Individual psychotherapeutic interventions in community-dwelling older adults, in turn, provides a personalized, one-on-one therapeutic environment where older adults can explore deep-seated issues under the direct guidance of a therapist [40]. This confidential setting fosters a strong therapeutic alliance, facilitating self-awareness and conflict resolution [17,40]. For instance, a recent study demonstrated the efficacy of individual cognitive behavioral therapy (CBT) in treating depressive symptoms among older adults, yielding significant improvements in symptoms and overall functioning [41]. The studies show that group therapy offers unique benefits that may be particularly advantageous for older adults struggling with depressive symptoms [37]. Group settings provide a supportive atmosphere where participants can relate to one another and share experiences meaningfully [42]. A meta-analysis showed that group therapy is effective in treating depressive symptoms among older adults, resulting in significant reductions in depressive symptoms and improvements in quality of life [43]. Another advantage of group therapy is its accessibility and cost-effectiveness. Conducted in a group setting, these interventions can reach a larger number of older adults, especially in communities where resources are limited [44]. While individual psychotherapeutic interventions may be effective for treating depressive symptoms in older adults, recent research suggests that group therapy offers greater efficacy [41,42,43,44]. One of the big reasons is that many professionals are not sufficiently trained to work with this type of population [45]. This fact is evident in different studies in which a lack of content on aging in different university degrees is evident. By providing a supportive environment, opportunities for shared experiences, and cost-effectiveness, group therapy presents a powerful approach to promoting recovery and enhancing the quality of life for older adults [40,43,44].

### 4.1. Strengths and Limitations of This Review

We have reported the review according to PRISMA recommendations and believe it has several methodological strengths. First, we pre-registered a protocol which is recommended to promote consistent conduct by the review team, and to ensure accountability, research integrity, and transparency of the eventual completed review. To ensure coverage of the relevant research, we conducted a search of literature databases without specifying which interventions we were looking for, as well as contacting experts in the field. We used four independent researchers to identify titles and abstracts of studies for inclusion and checked data extraction for errors. We assessed risk of bias and rated strength of evidence using recommended and widely used assessment tools. We used patient numbers rather than reported effect sizes in our analysis, attempting to obtain these from study authors when they were not reported. We then applied a consistent method of analysis for all included studies, meaning our estimates of effect were not biased by variations in the design of included trials. The fact that we include studies from several continents also seems to be a strength of our study. The main limitation was the non-inclusion of RCTs comparing psychotherapeutic interventions with psychopharmacology.

### 4.2. Implications for Future Research

Our study shows that the selected studies on group psychotherapeutic interventions in community-dwelling older adults had a larger effect size. They provide an environment of social support, reduce isolation, and promote connection with other group members, which can be especially important in older people facing loneliness and loss of social networks. Also important is the evidence of specialized clinical interventions with the older adult population to reduce levels of depressive symptoms and increase quality of life. In addition to this practical addition, the selection of the most appropriate clinical assessment tools to assess and monitor levels of depressive symptoms in older adults is something that we also clarify here. There are several validated assessment scales that are effective for this purpose; however, our study highlights the most suitable, widely accepted, and valid instruments for assessing and monitoring levels of depressive symptoms in older people.

## 5. Conclusions

In conclusion, both group and individual psychotherapeutic interventions effectively reduce depressive symptoms in community-dwelling older adults compared to control conditions, with group interventions showing greater overall efficacy. The reminiscence protocol and MBAT stand out for their significant impact. Group psychotherapeutic interventions in community-dwelling older adults provides a supportive environment where participants can connect with peers, share experiences, and combat loneliness, which are crucial factors in alleviating depressive symptoms among older adults. This approach also enhances socialization and offers a cost-effective solution, making it accessible to a broader population. Individual psychotherapeutic interventions such as the PROSPECT program and behavioral activation offer personalized support, prompting people to do activities they really enjoy. However, the communal benefits of group therapy, combined with its accessibility and cost-effectiveness, suggest it may be the superior approach for promoting recovery and improving the quality of life in older adults.

## Figures and Tables

**Figure 1 healthcare-12-02551-f001:**
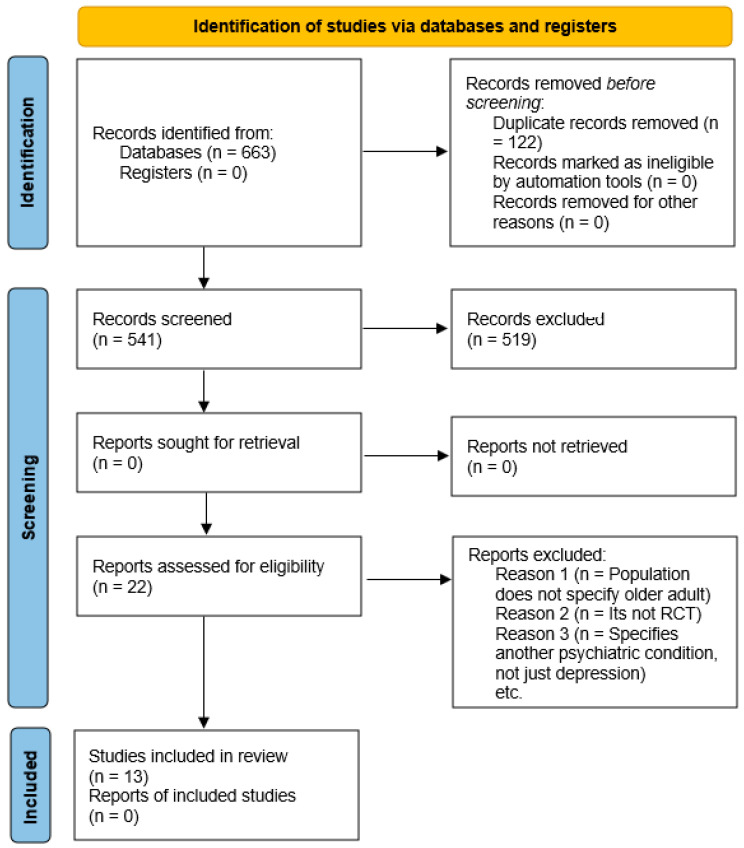
PRISMA flowchart.

**Figure 2 healthcare-12-02551-f002:**
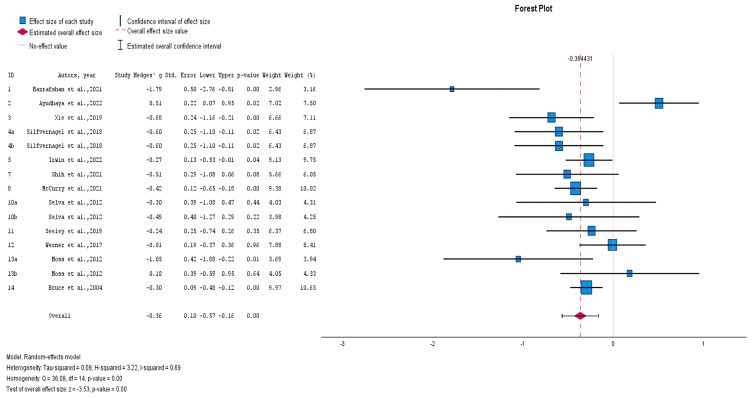
Effects of non-pharmacological interventions on depressive symptoms, post-intervention [23,25,26,27,28,29,30,32,33,34,35,36].

**Figure 3 healthcare-12-02551-f003:**
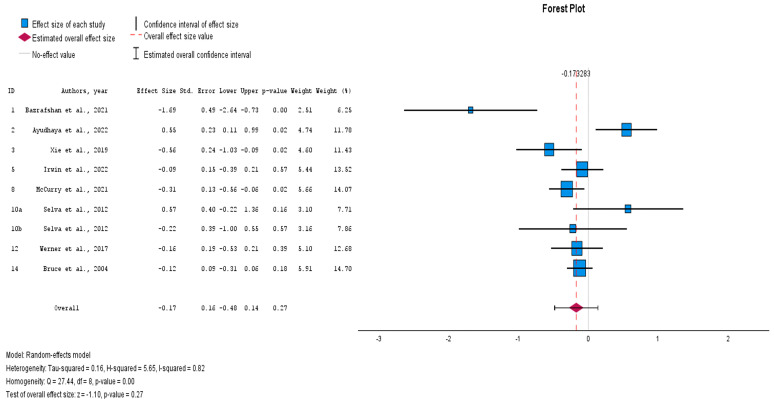
Effects of non-pharmacological interventions on depressive symptoms, follow-up [23,25,26,28,30,32,34,35].

**Figure 4 healthcare-12-02551-f004:**
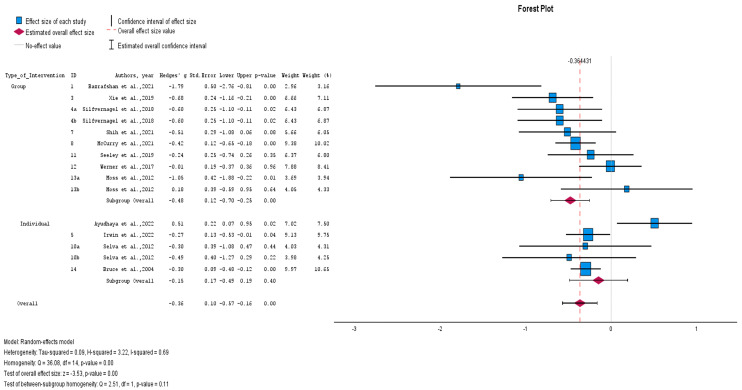
Effects of individual and group non-pharmacological interventions [23,25,26,27,28,29,30,32,33,34,35,36].

**Figure 5 healthcare-12-02551-f005:**
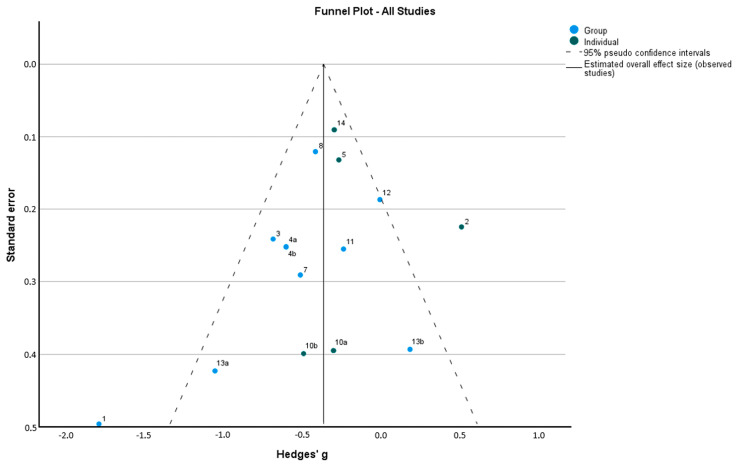
Funnel plot of all studies (Egger’s regression-based test) [23,25,26,27,28,29,30,32,33,34,35,36].

**Figure 6 healthcare-12-02551-f006:**
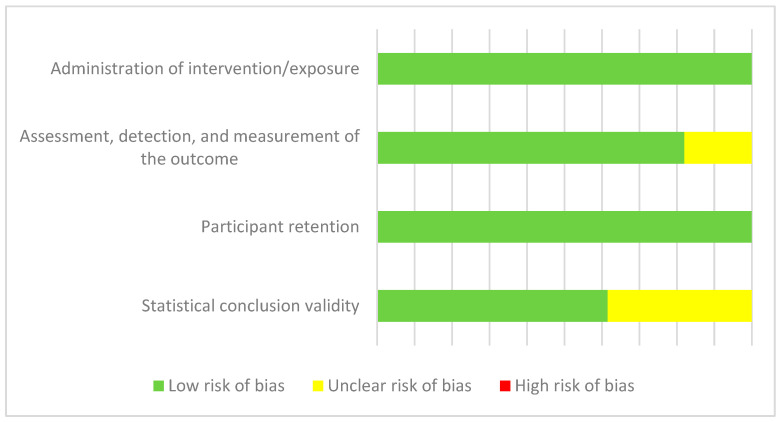
Assessment of the risk of bias (JBI).

**Figure 7 healthcare-12-02551-f007:**
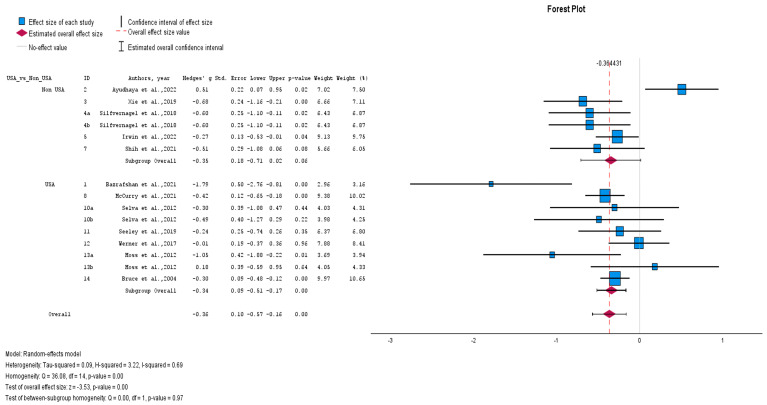
USA vs. non-USA subgroup analyses (forest plot) [23,25,26,27,28,29,30,32,33,34,35,36].

**Table 1 healthcare-12-02551-t001:** Study characteristics.

No.	Authors/Year	Sample	Aim	Results/ Conclusion	Country
1	[23]	24 female older adults who used the hookah	The aim of this study was to investigate the effect of structured group reminiscence therapy on depression and anxiety in older women who use a hookah	In the intervention group, a statistically significant difference in depression, anxiety, and the number of smoking daily in three stages (*p* < 0.05) was observed. Also, comparison of the intervention and control groups showed that the differences were significant (*p* < 0.05). Structured group reminiscence is an effective intervention used to reduce the symptoms of depression and anxiety and the amount of daily smoking among older women who use a hookah.	USA
2	[25]	82 older adults	This study aimed to evaluate the effectiveness of an adapted BA among older Thai adults with sub-threshold depression residing in the community. Program effectiveness was assessed using two objective measures, (a) daily step count as the indicator of daily (physical) activity level and (b) HRV indexes as the biomarkers of depressive symptoms.	The results suggested that behavioral activation (BA) may have a therapeutic effect on depression symptoms of older adults with subthreshold depression via improved heart rate variability (HRV).	Thailand
3	[26]	80 older adults	This study evaluated the effectiveness of a modified behavioral activation treatment (MBAT) intervention on reducing depressive symptoms in rural left-behind older adults.	MBAT produced a significantly greater reduction in depressive symptoms than regular care in rural left-behind older adults.	China
4	[27]	66 older adults	The aim of the study was to compare the effects of an eight-week-tailored internet-supported cognitive behavior therapy (iCBT) program and to compare it against the provision of weekly general support.	iCBT may be effective for some older adults and the role of cognitive function needs to be investigated further.	Switzerland
5	[28]	291 older adults	The aim was to examine whether treatment of insomnia disorder with cognitive behavioral therapy for insomnia (CBT-I) compared with an active comparator condition, sleep education therapy (SET), prevents major depressive disorder in older adults.	The findings of this randomized clinical trial indicate that treatment of insomnia with CBT-I has an overall benefit in the prevention of incident and recurrent major depression in older adults with insomnia disorder. Community-level screening for insomnia concerns in older adults and wide delivery of CBT-I–based treatment for insomnia could substantially advance public health efforts to treat insomnia and prevent depression in this vulnerable older adult population.	USA
6	[29]	57 older adults	This study aimed to examine the efficacy and cognitive mechanisms of MBCT in older adults with active depressive symptoms	Although both MBCT and active control program decrease the severity of depressive symptoms in older adults, only MBCT improved rumination and mindfulness. Older adults with more severe depression and more severe dysfunctional cognition may benefit more from the specific therapeutic effects of MBCT.	Hong Kong
7	[30]	327 older adults	To evaluate the effectiveness of telephone CBT-I vs. education-only control (EOC) in older adults with moderate to severe osteoarthritis pain.	In this randomized clinical trial, telephone CBT-I was effective in improving sleep, fatigue, and to a lesser degree, pain among older adults with comorbid insomnia and OA pain in a large statewide health plan. The results supported the provision of telephone CBT-I as an accessible, individualized, effective, and scalable insomnia treatment.	USA
8	[31]	81 older adults	This study assessed the effects of an autobiographical memory intervention on the prevention and reduction of depressive symptoms in older persons in residential care. Trained volunteers delivered the intervention.	Depressive symptoms improved equally in the intervention and the control condition at post measurement. Participants with clinically relevant depressive symptoms also maintained the effects at follow-up in both conditions. The retrieval of specific positive memories improved more in the autobiographical memory intervention, although this was not maintained at follow-up. Anxiety and loneliness improved equally well in both conditions, but no effects were found for well-being or mastery.	The Netherlands
9	[32]	37 older adults	The aim of this experiment was to examine the efficacy of life review based on autobiographical retrieval practice for treating depression in older adults	Results indicated decreased depression for both conditions, with no significant differences between the two therapies. There was some indication of greater gain in production of specific memories among those in life review therapy. Patients who produced higher numbers of specific memories decreased their depression scores at a faster rate.	USA
10	[33]	93 older adults	To assess the feasibility, acceptability, and potential effectiveness of a low-intensity, peer-supported cognitive behavioral intervention for mild to moderate depression and/or anxiety.	Clinically significant reduction in depressive symptoms in the experimental group compared to the control group, although the effect was not statistically significant.	USA
11	[34]	117 older adults	The aim of this study was to examine the effect of interactive group music therapy versus recreational group singing on depressive symptoms in older adults nursing home residents.	The results suggest that music therapy decreases depressive symptoms in older adults in nursing homes more effectively than recreational singing.	USA
12	[35]	598 older adults	The aim was to determine the effect of a “PROSPECT (prevention of suicide in primary care elderly: collaborative trial)” intervention in primary health care on suicidal ideation and depression in older people.	Rates of suicidal ideation decreased more rapidly in patients undergoing the intervention compared to patients under usual care. Patients undergoing the intervention had a more favorable evolution of depression, both in the degree and speed of symptom reduction. The effects on depression were not significant among patients with mild depression unless there was suicidal ideation.	USA
13	[36]	26 older adults	This study investigated behavioral activation (BA) bibliotherapy as a treatment for late-life depressive symptoms.	Because the study control was lost after the delayed treatment group received the intervention, within-subjects analyses examining both treatment groups combined showed that clinician-rated depressive symptoms significantly decreased from pre-treatment to both post-treatment and 1-month follow-up. Self-reported depressive symptoms were significantly lower from pre-treatment to 1-month follow-up. These findings suggest that BA may be useful in treating mild or subthreshold depressive symptoms in an older adult population.	USA

**Table 2 healthcare-12-02551-t002:** Clinical assessment instruments for assessing levels of depression.

No.	Clinical Assessment Tool	Included Studies
1	Patient Health Questionnaire-9 (PHQ-9)	[27,28,30,33]
2	Montgomery Åsberg Depression Rating Scale—Self Rated	[27,34]
3	Geriatric Depression Scale (GDS)	[26,32,36]
4	Elderly Depression Inventory	[23]
5	Thirty self-rated items of the Thai Geriatric Depression Scale	[25]
6	Hamilton Depression Rating Scale (HDRS)	[29,35,36]

**Table 3 healthcare-12-02551-t003:** Univariate and multivariate random effects meta-regression analyses.

Type	Moderator	*n*	Model Coefficient Test	Residual Heterogeneity	
			χ^2^	I^2^ (%)	R^2^ (%)
Categorical	Country	15	18.8 **	0.1	100
Continuous	Year	15	0	68.3	0
Continuous	Mean age	15	1.21	68	0.6
Continuous	Women Percent	15	1.06	71.7	0

OBS: **—*p* ≤ 0.001.

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
