# Peer review of "Psychotherapeutic Interventions for Depressive Symptoms in Community-Dwelling Older Adults: A Systematic Review with Meta-Analysis"

_healthcare, 2024, doi:10.3390/healthcare12242551_

Round 1

Reviewer 1 Report (Previous Reviewer 1)

Comments and Suggestions for Authors

It is essential that the manuscript includes a detailed report of the results from the quality assessment and risk of bias evaluation of the included studies. The lack of this information limits the reader’s ability to interpret the robustness and validity of the findings. I suggest incorporating a clear description of the methods used to assess study quality, as well as specific results for each study or group of studies. This would strengthen the credibility of the meta-analysis and provide a more comprehensive evaluation of the evidence.

The PRISMA flowchart provided in the manuscript does not fully align with current recommendations established by the PRISMA guidelines. Adhering to these guidelines would increase the transparency of the selection process and help readers better understand how the final set of included studies was determined.

Table 2 could benefit from additional details about the included studies. These details would facilitate the interpretation of the results and improve the understanding of each study’s context within the meta-analysis.

The forest plots represent critical information in a meta-analysis, and placing them in the supplementary material restricts readers' access to these key results. It is recommended to move the forest plots to the main text to allow for a more direct review of effect sizes and heterogeneity across studies.

The absence of a sensitivity analysis and subgroup analyses is an important limitation. Since the included studies may have used different instruments to assess depressive symptoms, conducting subgroup analyses by instrument would be useful for assessing the consistency of results. Additionally, a sensitivity analysis could help evaluate the influence of individual studies on the overall results, thus enhancing the robustness of the meta-analysis conclusions.

Author Response

  1. It is essential that the manuscript includes a detailed report of the results from the quality assessment and risk of bias evaluation of the included studies. The lack of this information limits the reader’s ability to interpret the robustness and validity of the findings. I suggest incorporating a clear description of the methods used to assess study quality, as well as specific results for each study or group of studies. This would strengthen the credibility of the meta-analysis and provide a more comprehensive evaluation of the evidence.

Authors: We appreciate your suggestion and have incorporated a clear description of the results used to assess study quality. Additionally, we have included specific results section called “Assessment of the risk of bias” and provide a more thorough evaluation of the evidence. Please see page 14, lines 348 to 360

  1. The PRISMA flowchart provided in the manuscript does not fully align with current recommendations established by the PRISMA guidelines. Adhering to these guidelines would increase the transparency of the selection process and help readers better understand how the final set of included studies was determined.

Authors: We updated the PRISMA Flowchart to the most up to date one in order to increase the transparency of the selection process and help readers better understand how the final set of included studies was determined.

  1. Table 2 could benefit from additional details about the included studies. These details would facilitate the interpretation of the results and improve the understanding of each study’s context within the meta-analysis.

Authors: Thank you for your valuable suggestion regarding Table 2. To address this, we have included a detailed description of the findings presented in the table within the main text of the manuscript. This addition aims to enhance the interpretability of the results and provide a clearer understanding. We believe that this textual explanation complements the information in Table 2, making it more accessible and enriching the overall analysis. Please see page 11, Lines 304 to 321.

  1. The forest plots represent critical information in a meta-analysis and placing them in the supplementary material restricts readers' access to these key results. It is recommended to move the forest plots to the main text to allow for a more direct review of effect sizes and heterogeneity across studies.

Authors: Thank you for your insightful feedback regarding the placement of the forest plots. In response to your recommendation, we have moved the forest plots from the supplementary material to the main text. This adjustment ensures that readers have direct access to these key results, facilitating a more thorough review of the effect sizes and heterogeneity across studies. Please see pages 12 to 13, Lines 324 to 347.

  1. The absence of a sensitivity analysis and subgroup analyses is an important limitation. Since the included studies may have used different instruments to assess depressive symptoms, conducting subgroup analyses by instrument would be useful for assessing the consistency of results. Additionally, a sensitivity analysis could help evaluate the influence of individual studies on the overall results, thus enhancing the robustness of the meta-analysis conclusions.

Authors: In response, we have added two new sections to the results: "Subgroup Analyses" and "Sensitivity Analysis", as well as a section titled "Moderator Analysis". These additions address the potential variability in instruments used to assess depressive symptoms, evaluate the influence of individual studies on the overall results, and explore potential moderators, thereby enhancing the robustness and comprehensiveness of our meta-analysis conclusions. Please see page 14 to 15, Lines 362 to 385.

Reviewer 2 Report (Previous Reviewer 3)

Comments and Suggestions for Authors

Dear authors,

The manuscript has improved substantially with the changes made.

Congratulations.

Kind regards.

Author Response

Dear authors,

The manuscript has improved substantially with the changes made.

Authors: Thank you for your positive feedback. We are delighted to hear that the manuscript has improved substantially with the changes made. Your guidance and insights were invaluable throughout the revision process.

Congratulations.

Kind regards.

Round 2

Reviewer 1 Report (Previous Reviewer 1)

Comments and Suggestions for Authors

The authors have improved this version and responded to comments.

This manuscript is a resubmission of an earlier submission. The following is a list of the peer review reports and author responses from that submission.

Round 1

Reviewer 1 Report

Comments and Suggestions for Authors

It would be valuable to include quantitative epidemiological data in the Introduction to better contextualize the magnitude of the problem studied. Additionally, there are minor writing errors, such as a missing period in the third paragraph.

I recommend ensuring that the cited references align with the content of the manuscript. For instance, in some cases, the references do not support the claims made by the authors. Also, in parts of the Introduction, “clinical practice guidelines” are mentioned in the plural, but only reference 9 is cited.

I do not find it appropriate to mention a preliminary search in PubMed in the Introduction. Also, the correct term is "PubMed," and the search as described does not seem suitable.

It’s important to avoid using "depression" and "depressive symptoms" interchangeably. The authors should follow the PRISMA guidelines for reporting systematic reviews.

I suggest including all search strategies and their results as supplementary material, as well as listing excluded studies with reasons for exclusion.

The authors should revise and improve their search strategies using appropriate MeSH terms. More details should be provided regarding who extracted the data and how conflicts were resolved.

It’s unclear whether the extracted data were means, relative risks, odds ratios, or absolute numbers. Furthermore, the authors should present forest plots for the meta-analysis.

The authors should also explain why the registered protocol mentioned that a meta-analysis would not be performed, but it was included in the manuscript.

Finally, it is important to expand on the limitations of the study to provide greater transparency about the results.

Comments on the Quality of English Language

Moderate editing of English language required.

Reviewer 2 Report

Comments and Suggestions for Authors

Thank you very much for the opportunity to review your paper entitled: Psychotherapeutic interventions for depressive symptoms in community-dwelling older adults: A systematic review with meta-analysis. This a well-written, clear article with a strong methodology.

Please find some suggestions below:

ABSTRACT

Please separate abstract and background.

Whats is the added value of MBAT in the abstract?

INTRODUCTION

Although depression is clear to many, it is useful to include a definition of depression so it is clear what definition is used in this study.

L64 Why do elderly not receive psychometric interventions?

L84 + L91 Please check the reference. E.g. you jump from [14] to [20].

L102-103 needs a reference.

L103 - 105 needs a reference.

L114 please rephrase (the aims aims to)

METHODS

L124-126 Please check this sentence (examining)

L126 Please mention the exact period instead of last 10 years. This way, it is clear to the reader what the period was.

L128 why putting RCTs between brackets? 

L128 - 134 needs a reference

L135 - 138 needs e reference

L222 (Joanna Briggs Institute quality assessment tool) needs a reference

RESULTS

Table 1

please be consistent (P<.05 - p<.05)

80 chenese older adults? Chinese? If so, why mentioning here 80 Chinese older adults, but in the previous you only mention 82 older adults instead of 82 older Thai adults.

4: we conclude that... the other conclusions are written more neutral.

7: 60 years and older. Is it useful to mention this as it is one of the inclusion criteria and it has not been mentioned previously.

10-11-12: another way of presenting the studygroup

DISCUSSION

Isn't it a strength you included articles from three different continents?

CONCLUSION

L376 in in

REFERENCES

4 and 5 are the same

8 American Psychological Association

Comments on the Quality of English Language

Clear language, some minor errors.

Reviewer 3 Report

Comments and Suggestions for Authors

Dear authors, congratulations on this research. Below I detail some suggestions:

1. Introduction:

-This part details other problems that the elderly population may present that also affect depression. However, cardiovascular problems, degenerative conditions, dementia and, in particular, cancer must be specifically mentioned, which significantly affects the psychosocial life of the person (Useche-Guerrero, et al., 2024. https://doi.org /10.1155/2024/6698804 ).

-Line 107: Why was it only searched in PUBMED? perhaps results could have been found in WOS or SCOPUS. 

2. Methods

-2.2. Review question. Formulate the review question in PICO format.

-Line 168: Provide bibliographical citation that supports this statement.

-2.4. Search strategy: Why were the WOS and Scopus databases not included? 

3. Results:

-The description of results must be improved. A subsection should be added in the results section that describes the general results (countries in which the studies included in the systematic review were carried out, designs of the included studies, types of population...). Afterwards, another subsection with specific results must be added. In this subsection, the most specific results related to the objective of the research must be commented on.

4. Discussion:

-Lines 340 - 343: An important problem for working with the elderly population must be added in this part. This statement will give more consistency to the discussion section. Many professionals are not sufficiently trained to work with this type of population. This fact is evident in different studies in which a lack of content on aging in different university degrees is evident (Moreno-Sánchez, E., Gago-Valiente, F.-J., Mendoza-Sierra, M.-I ., Sáez-Padilla, J., Castillo-Viera, E., Campos-de-Sousa-Faria, M.-C., Lozano-Carvalhal, S.-R., Justo-Henriques, S.-I., Martins-Texeira-Da-Costa, E.-I., Gouveia-Baltazar, M.-J., & Da-Costa-Bizarro-Morais, D.-M. (2024). a study in Spain and Portugal. Behavior Analysis and Modification, 50(182), 109-128).

Kind regards.